# Diversity of plant parasitic nematodes characterized from fields of the French national monitoring programme for the Columbia root-knot nematode

Nathan Garcia[1,2]*, Eric Grenier[2], Alain Buisson[1], Laurent Folcher[1]

**1** Plant Health Laboratory – Nematology Unit, French Agency for Food, Environmental and Occupational Health and Safety, Le Rheu, Ille et Vilaine, France, **2** Institute for Genetics, Environment and Plant Protection, French National Institute for Agricultural Research and Environment, Le Rheu, Ille et Vilaine, France

* nathan.garcia@anses.fr

**Data Availability Statement:** All relevant data are within the paper and its Supporting information files.

## Abstract

Plant parasitic nematodes are highly abundant in all agrosystems and some species can have a major impact on crop yields. To avoid the use of chemical agents and to find alternative methods to manage these pests, research studies have mainly focused on plant resistance genes and biocontrol methods involving host plants or natural enemies. A specific alternative method may consist in supporting non-damaging indigenous species that could compete with damaging introduced species to decrease and keep their abundance at low level. For this purpose, knowledge about the biodiversity, structure and functioning of these indigenous communities is needed in order to carry out better risk assessments and to develop possible future management strategies. Here, we investigated 35 root crop fields in eight regions over two consecutive years. The aims were to describe plant parasitic nematode diversity and to assess the potential effects of cultivation practices and environmental variables on communities. Community biodiversity included 10 taxa of plant parasitic nematodes. Despite no significant abundance variations between the two sampling years, structures of communities varied among the different regions. Metadata collected for the past six years, characterizing the cultural practices and soils properties, made it possible to evaluate the impact of these variables both on the whole community and on each taxon separately. Our results suggest that, at a large scale, many variables drive the structuration of the communities. Soil variables, but also rainfall, explain the population density variations among the geographical areas. The effect of the variables differed among the taxa, but fields with few herbicide applications and being pH neutral with low heavy metal and nitrogen concentrations had the highest plant parasitic nematode densities. We discuss how these variables can affect nematode communities either directly or indirectly. These types of studies can help to better understand the variables driving the nematode communities structuration in order to support the abundance of indigenous non-damaging communities that could compete with the invasive species.

**Funding:** This study is part of a PhD project, funded by the Brittany Regional Council - https://www.bretagne.bzh/ - and the French Agency for Food, Environmental and Occupational Health & Safety (ANSES) - https://www.anses.fr/en - The funders had no role in study design, data collection and analysis, decision to publish, or preparation of the manuscript.

**Competing interests:** The authors have declared that no competing interests exist.

## Introduction

Plant parasitic nematodes (PPNs) are soil organisms that feed on plant tissues. A limited number of these nematodes can induce economic losses by decreasing yield or causing visual damage, making the products unsuitable for sale [1]. Even though most PPNs are not damaging to crops at the abundances commonly found in agricultural soils, some taxa are known to have a major negative impact. Jones *et al.* [2] listed the 10 most important plant parasitic nematodes, among which *Meloidogyne* spp. (also called root-knot nematodes) are in first place. In particular, the Columbia root-knot nematode, *Meloidogyne chitwoodi*, represents a threat to temperate agriculture, mainly for root crops [2]. In order to manage this quarantine pest in fields where it has been detected, several studies have focused on non-host and resistant plants [3,4], or on biocontrol strategies [5]. However, studies focusing on preventive strategies, such as preventing the establishment of the damaging nematodes in fields where they are not yet present, are lacking.

The biodiversity of indigenous PPN communities can play a major role in the establishment success of a newly introduced species, since indigenous species could be potential antagonists to the introduced one. For instance, competition for roots between PPN species among communities, has been observed [6–9]. In particular, Garcia *et al.* [9] showed, in a greenhouse experiment, that the risk of establishment of *M. chitwoodi* increases in less diverse and less abundant PPN communities. As a result, knowledge about the biodiversity and factors affecting indigenous PPN communities in root crop fields is needed. By extension, such knowledge could provide an ecological risk assessment tool that could be useful to improve the choice of the fields to sample in the framework of the French programme for monitoring the Columbia root-knot nematode not only considering the susceptible host plant but also on the indigenous PPN communities.

In agro-ecosystems, that are shaped by frequent anthropogenic disruptions, the structure and functioning of PPN communities depend mainly on crop management practices [10–13], but also on environmental conditions [12–16]. Intensive profound tillage is probably the most commonly studied cultivation practice and it has been shown that this tillage practice usually has a negative impact on PPN populations [10,17]. Cultivation practices are known to change the physico-chemical properties of the soil [18,19], which also affects the nematodes in the soil, by modifying the metabolism, movement or food accessibility for example [13,14,16]. However, studies often focus on a few environmental conditions or crop practices at the same time (*e. g.*: profound tillage, soil texture, or heavy metal content) [11,14,15]. There are, to our knowledge, only a few studies dealing with a larger number of variables and/or at broad spatial scales [20,21]. In the present study, we sampled fields that are part of the French monitoring programme for *M. chitwoodi* in several geographical areas in order to (i) describe the diversity and abundance of PPN communities in fields including a root crop in the crop rotation, (ii) evaluate whether cultivation practices identified in the literature at smaller scales in fact shape PPN communities at larger scale and (iii) assess the potential impact of environmental variables on PPN communities, including climate and physico-chemical conditions of soil.

## Results

### Characterization of the environment and PPN biodiversity

The sum of monthly precipitation and mean monthly temperatures were collected for the 12 months before each sampling date. Comparison tests showed that the climate was not significantly different over the two sampling years (W = 4.86, *p*-value = 0.283 and W = 638, *p*-value = 0.430 for precipitation and temperature comparisons, respectively).

**Table 1. Abundance comparison of the PPN communities found in 2015 and 2016 samplings.**

| | Mean ± Standard error in 100g of soil | | Comparison 2015–2016 | | | Prevalence (%) | |
|---|---|---|---|---|---|---|---|
| | **2015 (N = 35)** | **2016 (N = 31)** | **W** | **p-value** | **significance** | **2015** | **2016** |
| *Helicotylenchus* | 26.2 ± 9.2 | 67.9 ± 23.0 | 498.0 | 0.537 | | 46 | 45 |
| *Pratylenchus* | 60.5 ± 10.4 | 67.3 ± 14.6 | 567.0 | 0.757 | | 91 | 77 |
| *Amplimerlinius* | 14.5 ± 5.8 | 16.5 ± 5.5 | 536.5 | 0.935 | | 37 | 35 |
| Other Telotylenchidae | 38.7 ± 6.7 | 84.9 ± 29.0 | 380.5 | 0.038 | * | 91 | 97 |
| *Paratylenchus* | 20.9 ± 6.7 | 38.3 ± 13.1 | 481.5 | 0.414 | | 54 | 55 |
| *Criconemoïdes* | 2.6 ± 1.2 | 5.0 ± 2.6 | 524.0 | 0.746 | | 20 | 23 |
| *Meloidogyne* | 13.4 ± 5.9 | 8.0 ± 3.0 | 608.0 | 0.353 | | 49 | 35 |
| *Ditylenchus* | 0.5 ± 0.4 | - | - | - | | 9 | 0 |
| *Trichodorus* | 1.3 ± 1.3 | 18.6 ± 6.0 | 325.0 | 2.09e$^{-4}$ | *** | 6 | 45 |
| *Heterodera* (juveniles & eggs) | 1312.6 ± 637.5 | 1685.6 ± 645.2 | 502.0 | 0.593 | | 54 | 61 |
| Shannon-Weaver Index | 0.356 ± 0.037 | 0.413 ± 0.036 | 733.5 | 0.231 | | | |

That data depict the mean of 35 and 31 fields surveyed in 2015 and 2016, respectively. "W" indicate the value of the Wilcoxon test;

* $< 0.05$;

** $< 0.005$;

*** $< 0.001$.

Prevalence indicate the percentage of fields where the taxon was found among all the fields of the same year.

For the 35 fields sampled in 2015, 10 taxa were identified and counted (9 genera and one family for which we were unable to identify nematodes to the genus level) (Table 1). PPNs identified were *Helicotylenchus*, *Pratylenchus*, *Amplimerlinius*, *Paratylenchus*, *Criconemoides*, *Meloidogyne*, *Ditylenchus*, *Trichodorus*, *Heterodera* and the Telotylenchidae family. Among these taxa, the most frequently found were *Pratylenchus* and Telotylenchidae, both identified in 91% of the fields (Table 1). *Ditylenchus* was found only in 3 fields (Table 1).

Only nine taxa from among those mentioned above were identified in the 2016 samples from the 31 fields surveyed (Table 1). No additional taxon was found in 2016 compared to 2015, but *Ditylenchus* was no longer found in 2016. *Pratylenchus* and Telotylenchidae were still the most frequently observed PPNs. A comparison between the 2015 and 2016 communities showed a significant difference in *Trichodorus* abundance, increasing from 1.3 ± 1.3 individuals per 100 g of fresh soil in 2015 to 18.6 ± 6.0 in 2016 (W = 325.0, *p*-value = 2.09e$^{-4}$) (Table 1). A significant difference was also observed between abundances of Telotylenchidae (increasing from 38.7 ± 6.7 individuals per 100 g of fresh soil to 84.9 ± 29.0, W = 380.5, *p*-value = 0.038). However, an overall PPN community comparison using the Shannon-Weaver diversity index, showed no significant differences between the 2015 and 2016 PPN communities (W = 733.5, *p*-value = 0.231).

Fields were sampled under the scope of the French monitoring program during the two consecutive years, no quarantine species was found in any of the samples and *Meloidogyne sp.* mentioned in this article correspond to non-quarantine species of *Meloidogyne*.

## Effects of the variables on the whole PPN communities

A total of 6 transformation-based redundancy analyses (tb-RDAs) were performed, implemented for each analysis the cultivation practices of the previous years, moving backwards (*i.e.* sequential accumulation of the cultural practices over the five years prior to the samplings). These analyses aim to assess the effects of cultural practices and environmental variables on the overall community and a potential cumulative effect of the cultivation practices over the

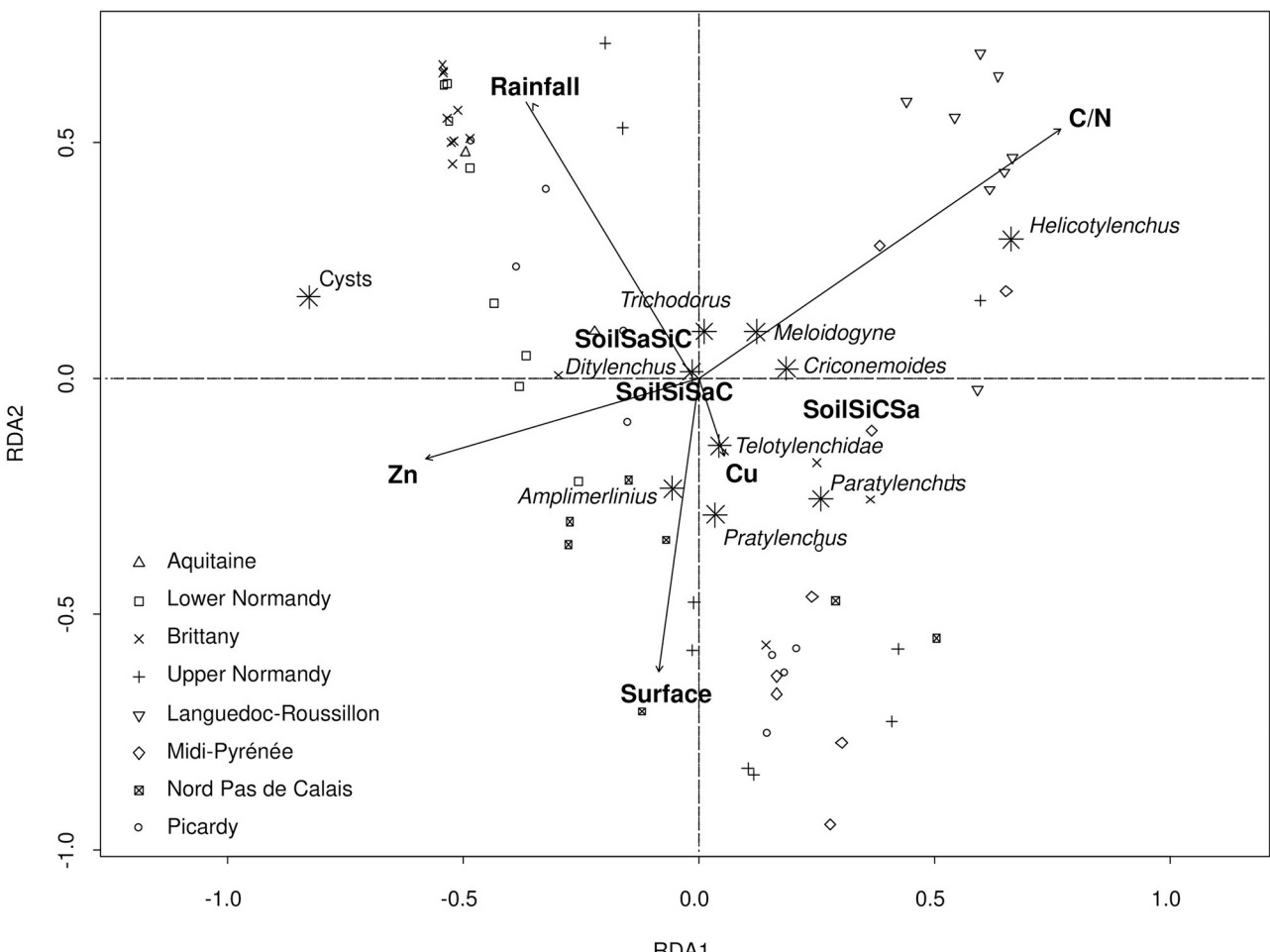

**Fig 1. Outputs of the tb-RDA for which the environmental variables and the cultivation practices of the sampling year have been implemented.**
Arrows and bold text indicate the significant variables after a permutation test. Modalities of soils depend on the proportion of sand, silt and clay in it (SoilSaSiC = majority of sand, then silt then clay; SoilSiCSa = majority of silt, than clay, then sand; SoilSiSaC = majority of silt, then sand, then clay).

5 past years (Figs 1 and S1). All variables were implemented to run each analysis without a preliminary selection step, since tb-RDA enables selection of only the significant variables with a permutation test. To ease the readability of the results, the first tb-RDA is presented here (Fig 1) (for which environmental variable and the cultural practices of the sampling year have been implemented) and the others tb-RDA (for which environmental variable and the cultural practices of the past years have been implemented) are available in supplementary data (S1 Fig).

The first two axes represented on Fig 1 explain 31% of the variance of the dataset. Environmental variables (mainly soil properties) appear to be the main drivers of PPN communities. Rainfall, C/N ratio, soil texture and heavy metal concentration (Zn or Cu) are retained on the analysis. The arrow length (indicating the importance of the effect of the variable) of the C/N ratio or rainfalls in particular indicates a key role in structuring of the communities (Fig 1). For example, *Helicotylenchus* abundance present higher abundance in communities found in soil with a higher C/N ratio. No agricultural practices appeared on this analysis (Fig 1) but a cumulative effect of non-herbicide application and profound tillage of the past years is observed on the following tb-RDA (S1 Fig).

The field surface area also appears on the analysis and we chose to use the surface as a random effect in the following analyses to take into account that the sampling protocol were not fully adapted to the field surface area.

Many PPN taxa are quite close to the center of the factorial map (Fig 1), indicating that the retained variables, despite a significant impact on overall community structure, have less importance for each taxon individually. As a consequence, we performed generalized linear mixed models (GLMMs) for each PPN taxon, in order to assess the impact of the same variables on the abundance variation between fields.

## Results of variable selection

Similarly to the tb-RDA, 6 multiple correspondence analyses (MCAs) were performed, adding the cultivation practices of the past years for each. The first two axes of each factorial map explained around 26% of the dataset variance. For the GLMM, we chose to retain only the variables showing at least two modalities on a minimum of two factorial maps. Hence, soil pH, Zn concentration in the soil, C/N ratio, and number of applications of herbicide products (present on 6, 5, 5 and 2 factorial maps, respectively were retained (factorial maps are available in supplementary data (S2 Fig)).

## Effect of the selected variables

After the variables selection step, we used a model-averaging approach based on GLMM to test the effective impact of the selected variables on the abundance variation of the PPN taxa. Similarly to the previous analyses, we ran several GLMMs to test a potential cumulative effect of the past applications of herbicide products on the PPN communities. We implemented the four selected variables in the GLMMs. Sixteen first-order models were ranked for each taxon and year or year period. The sum of weight (SW) and 95% confidence interval (CI) were calculated for the variables present in the subset of models with a $\Delta AICc < 6$. A summary of the results is presented in Table 2, while raw detailed results are available in supplementary material (S1 Table).

None of the considered variables impacted *Amplimerlinius* abundance in this study. One to three out of the four retained variables have appeared to significantly impact PPN abundance, depending on the taxa and the time period considered (Table 1). Soil pH–varying between 5.72 and 7.89 in our dataset–appears to positively impact the abundance of *Meloidogyne*, *Paratylenchus* and the Telotylenchidae while Zn concentration in the soil–ranging from 1.23 to 11.12 mg/kg–appears to negatively affect the abundance of *Trichodorus*, *Meloidogyne*, *Telotylenchidae* and *Pratylenchus*, for which the population abundance seemed to decrease in higher Zn concentration soils. However, Zn concentration appears to positively impact the abundance of *Heterodera*. Similarly, C/N ratio–ranging from 7.28 to 25.87 –appears to have a positive impact on *Meloidogyne*, *Criconemoides*, *Paratylenchus* and *Helicotylenchus*, but decrease *Pratylenchus* abundance. The application of herbicides, the only cultivation practice implemented in the GLMM following the MCA selection step, had a negative effect in the short or long term, depending on the taxon considered. PPN populations' abundance were lower in fields where herbicide products were often used. The results are unclear concerning the Telotylenchidae as herbicides had a negative impact if the sampling year (short-term) or the 5 years before the samplings (longest-term) were considered. However, herbicides had a positive impact on Telotylenchidae if cultivation practices of the years before the samplings, and 2 to 4 years before the sampling were considered.

**Table 2. Equations summarizing the model selection approach of the Poisson GLMMs for each taxa and each year periods.** Only significant variables are presented.

| Taxon | Year period | Significative variables | Taxon | Year period | Significative variables |
|---|---|---|---|---|---|
| *Helicotylenchus* | n | Helico ~ C/N - Herbi | *Criconemoides* | n | Crico ~ C/N - Herbi |
| | n-1 | Helico ~ - Herbi | | n-1 | Crico ~ - Herbi |
| | n-2 | Helico ~ C/N | | n-2 | Crico ~ C/N |
| | n-3 | Helico ~ - Herbi | | n-3 | Crico ~ C/N |
| | n-4 | Helico ~ - Herbi | | n-4 | Crico ~ C/N |
| | n-5 | Helico ~ C/N | | n-5 | Crico ~ C/N |
| *Pratylenchus* | n | Prat ~ - C/N - Zn | *Meloidogyne* | n | Melo ~ C/N + pH - Zn |
| | n-1 | Prat ~ - C/N - Zn - Herbi | | n-1 | Melo ~ C/N + pH - Zn |
| | n-2 | Prat ~ - C/N - Zn - Herbi | | n-2 | Melo ~ C/N + pH - Zn |
| | n-3 | Prat ~ - C/N - Zn - Herbi | | n-3 | Melo ~ C/N + pH - Zn |
| | n-4 | Prat ~ - C/N - Zn - Herbi | | n-4 | Melo ~ C/N + pH - Zn |
| | n-5 | Prat ~ - C/N - Herbi | | n-5 | Melo ~ C/N + pH - Zn |
| Telotylenchidae | n | Telotyl ~ pH - Zn - Herbi | *Trichodorus* | n | Tricho ~ - Zn - Herbi |
| | n-1 | Telotyl ~ pH - Zn + Herbi | | n-1 | Tricho ~ - Zn |
| | n-2 | Telotyl ~ pH - Zn + Herbi | | n-2 | Tricho ~ - Zn - Herbi |
| | n-3 | Telotyl ~ pH - Zn + Herbi | | n-3 | Tricho ~ - Zn |
| | n-4 | Telotyl ~ pH - Zn + Herbi | | n-4 | Tricho ~ - Zn - Herbi |
| | n-5 | Telotyl ~ - Herbi | | n-5 | Tricho ~ - Zn |
| *Paratylenchus* | n | Parat ~ C/N + pH | *Heterodera* | n | Hetero ~ Zn - Herbi |
| | n-1 | Parat ~ C/N + pH | | n-1 | Hetero ~ Zn |
| | n-2 | Parat ~ C/N + pH | | n-2 | Hetero ~ Zn - Herbi |
| | n-3 | Parat ~ C/N + pH | | n-3 | Hetero ~ Zn - Herbi |
| | n-4 | Parat ~ C/N + pH | | n-4 | Hetero ~ - Herbi |
| | n-5 | Parat ~ - Herbi | | n-5 | Hetero ~ Herbi |

"C/N" correspond to carbon/nitrogen ratio; "Herbi" correspond the number of applications of herbicide products over the time period considered; "Zn" correspond to the quantity of zinc in the soil; "Year period" indicate which year of cultural practices have been considered (*i.e.* "n" means the cultural practices of the sampling year, "n-1" means the cultural practices of the sampling year and the first past year etc.) to observe potential cumulative effect. No variables had a significant impact on *Amplimerlinus* population variations, explaining its absence in this table.

## Discussion

### PPN community biodiversity

This study describes the PPN communities in agrosystems including root crops in France. Sampling in eight geographical areas highlighted differences between communities due to environmental and cultural practices heterogeneity between the fields. The overall biodiversity, with 10 PPN taxa identified across the fields, is similar to other cereals and vegetables agrosystems studied in the literature [21–23]. The fields were chosen among those monitored as part of the French monitoring programme for the Columbia root-knot nematode, based on 2014 or 2015 crops (root vegetable crops), without information about crop rotation. It appears on the basis of crop management information collected from farmers that root crops were not very frequent in crop rotations, and farmers mainly grew wheat or maize, regardless of the region. Seven out of the ten taxa were present in at least 5 out of the 8 regions. The presence of the same PPN taxa in several regions could be due to their generalist feeding habits at the genus level and their plasticity regarding environmental conditions. Identified PPN taxa are mainly colonisers and are not particularly sensitive to environmental stress, explaining their presence in the majority of the regions [24,25]. However, one of the limitations here is that

despite the large geographic scale investigated, it was not possible to sample numerous fields in each region. It is therefore not possible to statistically compare the communities between regions more deeply. Interestingly, despite homogeneity between 2015 and 2016 PPN communities, *Trichodorus* was the only genus substantially more abundant in 2016. This nematode feeds ectoparasitically mainly on sugar beet and carrot, and can induce economic issues in sandy soil due to reduced plant growth known as "docking disorder" [1,26]. Beet or carrot was grown in 2015 in the fields with high 2016 abundance of *Trichodorus*, whereas crops cultivated before 2015 were generally cereals. The presence of a suitable host plant could explain this increase in abundance. Furthermore, *Trichodorus* has a very large host range [27], including many weeds that could have been present in sampled fields despite the use of herbicide chemicals. Similarly, the significant increase in Telotylenchidae between the two sampling years could be due to the presence of a better host plant in 2016, whether the crop grown that year or weeds. The other PPN taxa were not significantly different between the two years, probably because of short crop rotations dominated by cereals.

## The effects of variables on the overall PPN communities

In this study, we have observed mainly the impact of soil variables on the abundance variation of the PPN communities. Soil physico-chemicals properties, are known to be important factors for PPNs as they may modify their habitat, metabolisms, or movement for example [16,28] that could explain the highlighted effect of soil texture or rainfall. Heavy metals in soil such as Zn or Cu can shape PPN communities as these substances appeared to differentially decreased the abundance of PPNs in soil [14] as observed in this study for *Pratylenchus*, *Trichodorus*, *Meloidogyne* and Telotyenchidae, shown to be negatively correlated to Zn concentrations. The toxicity of heavy metals has been studied on growing media in *Caenorhabditis elegans*, a free-living nematode, and high concentrations have been shown to drastically increase mortality in both adult and juveniles [29,30], but also to reduce the growth of the nematodes, particularly in the descendants of parents exposed to heavy metals [30]. Heavy metals also alter the movement behaviour of the intoxicated individuals, possibly by attacking the muscles and the neural network [30]. At the community level studied here, little information is available about PPNs but heavy metals have an adverse impact on free-living soil nematode communities, thus including PPNs except cyst nematodes, by decreasing population abundances and community richness and modifying community structure [31,32]. Consequently, nematodes have developed a broad range of behavioural and physiological mechanisms (summarised in Ekschmitt and Korthals [14]) enabling avoidance, tolerance or detoxification of various heavy metals. The effects of heavy metals highlighted in this study can also be indirect through reduced plant growth [33] and thus lower quality nutritional content for PPNs, as these organisms depend on their host plants for nutrition.

Soil texture that appear in Fig 1 have also been shown in the literature to modify PPN community structure depending on the type of soil and particle size as they modify communities abundance and richness [13,16]. Sand, clay and silt proportions may affect the water retention but also the amount of minerals and organic matter present in soils [16,34]. This will impact plant growth and thus food quality for PPNs, but also movements through the water film around the soil particles, stimulated by the retention of root exudates that enable nematode to locate the roots [35], explaining the impact of soil texture on the communities sampled for this study.

Obviously, climate was a significant macro-variable affecting the PPN communities studied here, especially rainfall. Even at a larger scale (worldwide) Nielsen *et al*. [36] also observed an impact of rainfall on PPN communities. According to their observations, rainfall appears to be

the main factor affecting nematode family composition (including non-PPNs) by influencing local soil properties and vegetation that drive the nematode communities' structure. Temperature appears to be more important for PPNs in particular that appear the most abundant in warm locations, probably because of more stable ecosystems and abundant food resources than in cold location [36]. It is possible that in our study, soil variables were more important than climate compared to a worldwide scale, as climate variations in France, according to the world map of the Köppen-Geiger climate classification, are low and all the fields of this study were sampled in temperate regions with warm summers [37]. Precipitation, and thus moisture, increases plant growth, facilitates PPN movement in soil [38], and favours the PPN life cycle, for instance by increasing egg hatching [39].

Effect of field surface area on the PPNs is difficult to explain and could be an artefact. The sampling protocol were not fully adapted to the surface sampled (*i.e.* we did not sample more samples in the larger fields), but sampled seven soil cores over the longest diagonal in all fields. It therefore seems unlikely that the surface area affected PPN communities. More probably, field surface area reflects a characteristic of geographical regions as largest fields were sampled in Midi-Pyrénées and smallest fields were sampled in Nord-Pas-de-Calais.

Certain cultivation practices (e.g. tillage practice and application of non-herbicide products) also seem to play a key role for the PPN community's structure in the fields, even though these practices did not impact the abundance of the PPN taxa individually. Profound tillage, for example, may impact food accessibility for PPNs, by removing weeds, as well as their living habitats, mainly their living depth and the soil texture, as individuals are moved deeply in the soil [40,41]. Furthermore, some PPNs are less sensitive to environmental changes as reported for *Paratylenchus* [24] and thus may benefit from the soil disturbance compared to the other PPNs. Non-herbicide plant protection products (*i.e.* insecticides, fungicides or molluscicides) may eliminate potential natural enemies of PPNs such as nematophagous fungi [42], but they also have a negative impact on non-target organisms such as the PPNs [43], explaining their structuring role on PPN communities.

## Impact of the variables on each PPN taxon

Effects of the cultural practices and environmental variables were also tested on each PPN taxa using GLMMs. In our data, crop rotations were not retained in the GLMMs analyses after the dataset description using MCAs. This contrasts with results in the literature for which crops are known to be important for PPN community structure as richness and abundance vary with the crop rotations [44,45]. In our study, the impact of crop rotations may be masked by the environmental heterogeneity as at this country scale, environmental conditions (mainly soil) are much more diverse than the cultivation practices between the regions. Furthermore, and despite the presence of a root crop in the rotations, the dominance of cereals in all sampled fields, regardless of the region, probably homogenised the PPN communities.

We observed that herbicides have a significant negative impact, at least on long-term use, on the abundance of several PPN taxa (*Pratylenchus*, *Trichodorus*, *Helicotylenchus*, *Paratylenchus*, *Criconemoides*, *Heterodera* or Telotylenchidae), as has been reported previously [21,46,47]. It is not possible to rule out a potential direct effect of the products depending on the dose used [48], but it is more likely to be an indirect effect by removal of potential host weeds. Positive effects of herbicides can be observed on Telotylenchidae for accumulation of applications from years n-1 to n-4 (Table 2), possibly because of a negative impact on the other PPNs. Unidentified species among this family could be more competitive on the few remaining weeds. However, this interpretation is speculative, as the weed compositions, as well as potential intercrops between each year, were not assessed here.

In this study, soil physico-chemical variables (pH, soil C/N ratio and Zn) appears to be the main drivers of the PPN abundance variations for the identified taxa. The C/N ratio reflects organic matter and nitrogen enrichment. These variables generally characterise fertilizers to increase yields. As such, a positive effect may due to more dense root systems that are more convenient for PPNs development in the most fertilized fields. However, high nitrogen inputs for example have also been shown to be deleterious for soil nematodes because of the nematicidal effect of ammonia [15,49]. This was observed here because in the fields sampled for this study, the C/N ratios appear to be relatively low (<10)–except in the four fields sampled in Languedoc-Roussillon–and PPN abundances appear to be lower in soil with lower C/N ratios. With our dataset, Zn appears to negatively impact the abundance of *Trichodorus*, *Meloidogyne*, *Pratylenchus* and Telotylenchidae, confirming the results observed with the tb-RDA analyses for the whole PPN community.

## Conclusions

Based on our results at the spatial scale of several regions of France, it appears that soil and climate conditions are more important than cultivation practices for PPNs abundance, and these conditions may mask the effects of certain cultivation practices. Heavy metals, soil pH, and rainfall deeply modify PPN communities, affecting their environment and access to—or quality of—food resources. We showed, with 35 sampled fields in eight different geographical regions, that local climate and soil variations, but also certain cultivation practices, could explain the PPN community variations, despite similar diversity. However, it should be emphasized that anthropogenic activities, mainly in agrosystems, completely change the soil conditions, with high inputs, or tillage practice, which directly or indirectly have an impact on PPNs.

Even in the similar agrosystems sampled here, PPN communities may vary and it is probable that the differences in diversity would be higher in more different crop rotations. In this case, we can hypothesise that PPN communities in the different agrosystems would not have an equal resilience capacity to introduction of a new species, such as *Meloidogyne chitwoodi*, and an equal capacity to limit its development. It has been shown previously in greenhouse experiments that more diverse communities, especially those with high overall abundance of PPNs, regardless of the species, are able to limit the establishment of *M. chitwoodi* [9]. Given these results, it seems important to consider herbicide use, nitrogen inputs, and soil physico-chemical composition, as well as the cultivated crop, to select the fields sampled for the French national monitoring programme for the Columbia root-knot nematode. Further investigations are also needed to assess the potential of the different natural communities, which here do not damage crops, to control and limit the development of frequently introduced species, with the aim of proposing alternatives to pesticide use.

## Methods

### Ethics statement

No specific permits were required for the samplings. Permissions for sampling the fields were granted by the farmers (landowners). Furthermore, the 35 fields sampled in this study were also part of the 2015 French national monitoring programme for the Columbia root-knot nematode. The lands are not protected in any way and no protected plant or animal was sampled.

## Sampling and characteristics of the studied areas

Sampling was carried out from June to September 2015 in the same fields as those sampled in the framework of the national monitoring program dedicated to *Meloidogyne chitwoodi* and *M. fallax*. For this program, fields are chosen randomly among those that have grown a susceptible crop to *M. chitwoodi*, mainly root crops such as carrots, potatoes or beets. Eight regions were investigated: Aquitaine, Brittany, Languedoc-Roussillon, Midi-Pyrénées, Nord-Pas-de-Calais, Upper and Lower Normandy and Picardy. Sampling was repeated in 2016 in the same fields except three fields in Brittany and one field in Nord-Pas-de-Calais for which farmers did not allow us to sample again. All these fields are present in geographical areas defined as temperate regions with warm summers [37]. Mean monthly temperatures and rainfall data were collected from the closest meteorological station from "Terre-net.fr", an agricultural data website [50] monitoring climate in fields in France, for each sampled field over the year before the sampling date.

A total of 35 fields, ranging from 0.13 ha to 6.64 ha and representing 31 farms were sampled in 2015, and 31 of them were sampled again in 2016 as farmers from the other four fields (3 fields from Brittany and 1 field from Nord-Pas de Calais) did not complete the survey. For each field, information about cultivation practices from 2010 to 2016 was collected from farmers through a survey: crop rotations, type and number of tillage practices (profound tillage at 30 cm or deeper and superficial at less than 30 cm deep), and number of applications of plant protection products (herbicides or non-herbicides (including fungicides, insecticides and molluscicides). The physico-chemical properties of soils, including pH, organic C/N ratio, soil texture and Cu, Zn and Fe quantities, were obtained from "BDAT", a public database of the French National Institute for Agricultural Research and Environment [51]. Soil samples (around 1.5 kg) were composed of seven elementary soil cores taken at 30 cm depth along the longest diagonal of each field with a manual auger (diameter 2.5 cm). The GPS coordinates were recorded at each of the seven sample points in 2015 in order to repeat the same sampling in 2016.

## Nematode extraction and identification

The extraction protocol was the same as in Garcia *et al.* [21]. Briefly, mobile stages were extracted from 300 mL of fresh soil according to EPPO protocol for nematode extraction [52], using an Oostenbrink elutriator (Meku, Germany). Extraction was followed by two centrifugations (Hettich Rotanta 460, Germany), the first one with water and the second one with $MgSO_4$ with a density of 1.18 [52], to purify the nematode suspension. PPN genera were identified and counted in 5 mL of the suspension after a dilution step depending on the quantity of nematodes in the total suspension. Identifications were based on morphological criteria using a stereomicroscope or a microscope when more precision was needed [53,54]. Cysts were manually isolated from 500 mL of fresh soil under stereomicroscope after an adapted Baunacke methods on two successive sieves (800μm and 170μm) [52]. Genera were identified based on cyst morphology. Furthermore, the number of cysts was transformed into the number of eggs and juveniles per cyst, through extrapolation after crushing thirty-three cysts, to enable abundance comparisons with the other PPN taxa.

## Statistical analysis

For this study, we considered 15 different variables, both quantitative and qualitative: year of sampling, mean monthly rainfall, mean monthly temperature, C/N ratio, soil texture, soil Cu, Zn and Fe concentrations, soil pH, field surface area, crop rotation, number of superficial and profound tillage practices, number of herbicide and non-herbicide products used.

All statistical analyses were conducted using R software [55]. The Shannon index of species diversity was calculated for each PPN community as detailed in Spellerberg and Fedor (2003) [56]. A Wilcoxon test was used to assess differences between PPN taxa abundance and Shannon index in 2015 and 2016.

As samples were taken in quite different areas (mainly regarding environmental conditions), we wanted to assess the impact of both cultivation and environmental variables on the whole PPN community. Transformation-based redundancy analysis (tb-RDA) was performed using the "Vegan" R-package [57]. Since our dataset contained many zeros and according to Legendre and Gallagher (2001) [58] and Legendre and Legendre (2012) [59], we transformed our community data using Hellinger transformation [60] to allow the use of RDA (that is based on Euclidean distance, very sensitive to community matrices containing many zeros). Furthermore, the significance of the most contributive variables, was tested according to the Akaike information criterion (AIC) after both forward and backward selection with permutation tests. We performed six tb-RDAs to assess the impact of the past cultivation practices on the present PPN communities. For each analysis, the cultivation practices of the previous years were implemented one by one, moving backward (e.g. practices of year n for the first tb-RDA, practices of years n and n-1 for the second, practices of the years n, n-1 and n-2 for the third *etc.*), in order to evaluate whether accumulation of tillage practice or pesticides over the years could reveal a cumulative effect.

The effects of the variables on the abundance variation of each PPN taxon were evaluated, using a statistical strategy similar to that described by Garcia *et al.* [21]. Briefly, we first used Multiple Correspondence Analysis (MCA) using the "FactoMineR" R-package [61] to describe our dataset and to select the main contributing variables without any a priori knowledge [62,63]. MCA enables implementation of both qualitative and quantitative variables after grouping the values of the quantitative variables into two to four classes, in a way as balanced as possible to avoid giving greater weight to one modality compared to the others. Limits of each classes were the first and third quartiles and the median of the distribution of each variable. The PPN abundances were considered as supplementary variables and only the most contributive classes or modalities of variables (with an absolute contribution higher than twice the mean absolute contribution [64,65]) were kept on the factorial maps. Six MCA analyses were also performed, adding for each, the cultivation practices of the past years similarly to the tb-RDAs. We retained the variables for which two modalities at least were present on not less than two factorial maps. Then, the effective relationships between these selected variables and the abundance of the PPN taxa were tested using a model-averaging approach through the corrected Akaike information criterion (AICc) [62,63] on generalised linear mixed Poisson models (GLMMs) [66]. We used the "lme4" R-package [67] and the "MuMIn" R-package [68]. We considered the field surface area as a random variable since the sampling protocol was not adapted to the field surface area. According to Grueber *et al.* [63], we transformed our explanatory variables using Gelman's (2008) [69] approach, using the "arm" R-package [70], when the models failed to converge. The Sum of Weight (SW) and 95% Confident Interval (CI) were calculated for the variables present in the subset of models with a $\Delta$AICc < 6. Furthermore, if the 95% CIs included zero, the effect of the variables was considered uncertain [63] and thus not kept for the final model.

## Supporting information

**S1 Fig. Outputs of the tb-RDAs for each year period of cultivation practices.** Arrows and bold texts indicate the significant variables after a permutation test. Modalities of soils depend on the proportion of sand, silt and clay in it (SoilSaSiC = majority of sand, then silt then clay;

SoilSiCSa = majority of silt, than clay, then sand; SoilSiSaC = majority of silt, then sand, then clay)A: cultivation practices of year n and n-1 are implemented, the first two axes presented explain 38% of the dataset variance; B: cultivation practices of the n to n-2 are implemented, the first two axes presented explain 36% of the dataset variance; C: cultivation practices of year n to n-3 are implemented, the first two axes presented explain 35% of the dataset variance; D: cultivation practices of year n to n-4 are implemented, the first two axes presented explain 39% of the dataset variance; E: cultivation practices of year n to n-5 are implemented, the first two axes presented explain 36% of the dataset variance.
(TIF)

**S2 Fig. Outputs of the MCAs for each year period of cultivation practices.** Only the most contributing variables are projected on the maps. PPNs do not contribute to the factorial map construction. Classes for each variable are based on first quartile, median and third quartile. Abbreviations are presented in the supplementary material S2 Table. A: only cultivation practices of year n (year of sampling) are implemented in the analysis; B: cultivation practices of year n and n-1 are implemented; C: cultivation practices of year n to n-2 are implemented; D: cultivation practices of year n to n-3 are implemented; E: cultivation practices of year n to n-4 are implemented; F: cultivation practices of year n to n-5 are implemented.
(TIF)

**S1 Table. Raw model-averaging results for each taxon and year periods.** For each taxon, variable and year period considered, estimate, after the model averaging approach, 95% confident interval (95% CI) and the sum of weight (Σωi) are presented for the variables retain after the selection step.
(XLSX)

**S2 Table. Detail of the abbreviations used in MCA analysis outputs.**
(XLSX)

**S3 Table. Complete dataset (PPN community and environmental and cultural variables) for each year of sampling.**
(XLSX)

## Acknowledgments

We would like to thank all the farmers who allowed us to sample PPNs in their fields and the French Regional Services for Food for their help contacting the farmers. Thanks are also due to all the colleagues who gave their time to help sample, identify and count PPNs and discuss the results: Valentin Boulenger, Alain Buisson, Magali Esquibet, Sylvain Fournet, Didier Fouville, Josselin Montarry, Christophe Piriou, Catherine Porte, Elsa Ruliat, and Corinne Sarniguet.

## Author Contributions

**Conceptualization:** Nathan Garcia, Eric Grenier, Alain Buisson, Laurent Folcher.

**Data curation:** Nathan Garcia.

**Formal analysis:** Nathan Garcia.

**Funding acquisition:** Eric Grenier, Laurent Folcher.

**Investigation:** Nathan Garcia, Eric Grenier, Alain Buisson, Laurent Folcher.

**Methodology:** Nathan Garcia, Eric Grenier, Alain Buisson, Laurent Folcher.

**Project administration:** Nathan Garcia, Eric Grenier, Laurent Folcher.

**Resources:** Alain Buisson.

**Supervision:** Eric Grenier, Laurent Folcher.

**Validation:** Eric Grenier, Laurent Folcher.

**Writing – original draft:** Nathan Garcia.

**Writing – review & editing:** Nathan Garcia, Eric Grenier, Alain Buisson, Laurent Folcher.

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
