## [Decision Letter · Decision Letter 0]

23 Jul 2021

PONE-D-21-19128

Diversity of plant parasitic nematodes characterized from fields of the French national monitoring programme for the Columbia root-knot nematode

PLOS ONE

Dear Dr. Garcia,

Thank you for submitting your manuscript to PLOS ONE. After careful consideration, we feel that it has merit but does not fully meet PLOS ONE’s publication criteria as it currently stands. Therefore, we invite you to submit a revised version of the manuscript that addresses the points raised during the review process.

I appreciate your objective to explore the significance of various biotic and abiotic parameters for the distribution of plant parasitic nematode taxa, as we definitely lack understanding on this matter. That said, both reviewers raise a number of critical points, to which I agree. In its present form, your manuscript is not acceptable for publication, and I will only be able to assess its quality after substantial revision, where you carefully address all comments and points raised by the reviewers.

We look forward to receiving your revised manuscript.

Kind regards,

Mette Vestergård, Ph.D.

Academic Editor

PLOS ONE

Additional Editor Comments:

I appreciate your objective to explore the significance of various biotic and abiotic parameters for the distribution of plant parasitic nematode taxa, as we definitely lack understanding on this matter.

That said, both reviewers raise a number of critical points, to which I agree. In its present form, your manuscript is not acceptable for publication, and I will only be able to assess its quality after substantial revision, where you carefully address all comments and points raised by the reviewers.

Journal Requirements:

2. We noted in your submission details that a portion of your manuscript may have been presented or published elsewhere. [This research article is linked to the 2nd chapter of my PhD thesis, defended at the end of 2017, available on the HAL open archive (it is mandatory for French Ph.D. students), but never published in a peer-reviewed journal. The data, the analyses and the results presented are not strictly the same as those from my Ph.D. thesis manuscript and majority of the text has been reformulated.] Please clarify whether this [conference proceeding or publication] was peer-reviewed and formally published. If this work was previously peer-reviewed and published, in the cover letter please provide the reason that this work does not constitute dual publication and should be included in the current manuscript.

Reviewers' comments:

Reviewer's Responses to Questions

**Comments to the Author**

1. Is the manuscript technically sound, and do the data support the conclusions?

Reviewer #1: Partly

Reviewer #2: Partly

2. Has the statistical analysis been performed appropriately and rigorously? 

Reviewer #1: N/A

Reviewer #2: Yes

3. Have the authors made all data underlying the findings in their manuscript fully available?

Reviewer #1: Yes

Reviewer #2: Yes

4. Is the manuscript presented in an intelligible fashion and written in standard English?

Reviewer #1: Yes

Reviewer #2: No

5. Review Comments to the Author

Reviewer #1: The manuscript describes the effect of biotic and abiotic parameters on plant parasitic nematode communities and raises the question, if certain parameters can be used in the future to select "high-risk" fields for the detection of certain quarantine nematodes in monitoring programs. Unfortunatley, this hypothesis could not be addressed as no quarantine nematodes were detected. Besides, in praxis it is much easier and straight-forward to select fields based on good host plants, so the benefit of using plant parasitic nematode communities is not clear. If nematode communites are used to characterize certain scenarious, non-plant parasitic nematodes play an important role. However, this group of nematodes was completley ignored in the present study. It should at least be mentioned why non-plant parasitic nematodes were not considered in the study. Besides, there are numerous shortcomings and missing information as indicated in the separate file. Finding approriate answers to the raised question will decide, if the manuscript will finally be acceptabe or not.

Reviewer #2: NOTES FOR THE AUTHORS:

This is a novel study, and potentially a good addition to the literature that characterizes the ecology of plant-parasitic nematodes, however I would recommend publication only if major revisions are undertaken. The presentation of the manuscript is not yet of publication quality either by the standards in the field or by the expectations of the journal. Arguments and discussion are not fully developed, nor are they detailed, and frequently the reader is left attempting to follow a line of reasoning that relies entirely upon a necessity to read the cited references. There is heavy use of vague and/or subjective language (“various”, “several”, “important factors”, “major”, “more or less”, “certain groups”, “few”, “seems to vary”, “numerous”, “better quality”, etc.) which requires further explanation. Subjective language is used to describe changes or degrees of effect; this should be replaced with quantitative values or neutral language. Furthermore, the modelling and conclusions in this study are heavily reliant upon soil metadata that was not measured for this study, but retrieved from a database, potentially rendering these conclusions invalid if the metadata has changed.

INTRODUCTION:

General note:

Frequently, explanations and lines of argument are neither clear nor detailed, and often the reader is left attempting to follow a line of reasoning that has large gaps. For example, if a statement is made that cultivation practices or environmental conditions affect nematodes, this statement needs to be followed with explanatory text describing the specific types of disturbances that affect nematodes, and what the effects on the nematodes or nematode populations are.

Line 44: Technically unicellular organisms are considered to be micro-organisms, whereas nematodes (multicellular) are considered microfauna or mesofauna. Please adjust this wording for clarity and consistency with the field.

Line 47: Does “natural population levels” refer to crop population abundances or nematode abundances? Please specify.

Lines 55-57: “…could be potential antagonists.” This sentence is unclear, and the implication is vague; please specify what the introduced species could be antagonists for or to, and how the antagonism might occur.

Line 70: The term “soil work” is a non-standard term that is used frequently throughout this manuscript in reference to cultivation practices. In the interests of clarity, please use the standard terms “tillage practices” or “cultivation practices”, and further detail as necessary.

Line 72: Please describe how cultivation and physico-chemical properties of the soil affects nematodes. When creating a line of reasoning it is not sufficient to simply state that there is an effect, the effect should be described.

Line 73: Please list types of environmental conditions and crop practices that have been studied.

Lines 73-74: Please describe the nature of restricted areas and limited experimental designs.

Line 78: Please remind us of the importance of a root crop vs. a cereal crop.

RESULTS:

Lines 104, 106: Please specify whether per 100 g soil refers to 100 g wet (or field-moist) soil or dry soil?

Lines 107-109: I did not see any information in the Methods about performing calculations for the Shannon Index. Please include methodology, reference(s), and data.

Lines 110-112. This sentence is unclear. What does this mean?: “…and Meloidogyne sp. mentioned in the article correspond to other species.”

Line 136: Misspelling of “this”. “…cumulative effect of few practices…”; please list the practices to which you refer. Also, it is mentioned that agricultural practices were not retained in the analysis, and the sentence immediately contradicts itself by stating that effects due to practices were observed. Could this entire sentence be clarified?

Line 138: Please specify what you mean by a field surface. Whether or not a result is difficult to explain it should be listed in the results; further explanation can be done in the discussion, and if indeed the explanation is due to analysis artifacts, further explanation of the nature of the artifact is needed.

Lines 139-140: The methodology as described in this sentence appears to match sampling protocols outlined in the Methods section, thus contradicting the opening of the sentence. Please describe in detail what was not adaptable for this particular sampling.

Line 143: Please reference the appropriate figure or table after “…factorial map.”

Line 144: “seem” is subjective; simply list the impact of the variables quantitatively or statistically rather than rendering a judgment call as to perceived importance. Commentary of that nature should be moved to the discussion.

Line 144: By “structuration” do you mean “community structure”? Structuration is not the standardly accepted term for community structure, nor is it commonly used to refer to the formation or change of community structure.

Lines 167-182: The definitive statements “None of these variables impacted…”, “…had a positive impact…”, “…had a negative impact…” cannot be made. Remember that this study is largely statistical, and that that metadata originates from soil data that has not been measured in the laboratory for this study. At most it is possible to state that “…have appeared to positively/negatively impact…”. This paragraph requires significant clarification.

DISCUSSION:

General: There is frequent use of vague and subjective language (“various”, “several”, “seems”, “important factors”, “major”, “more or less”, “certain groups”, etc.) throughout this section. Please remove all subjective language and reframe in terms of quantitative and/or statistical values. Vague terminology requires clarification in the text, rather than following with a citation so that the audience is required to read the citation in order to draw their own conclusions.

Lines 187-188: “Various” is vague usage. Please specify the geographical areas and crop management approaches to which you refer.

Line 189: What do you mean by “close to that of other similar agrosystems”? “Close” is subjective; what of your data supports this assertion, and please be very specific in your response. What are “similar” agrosystems?

Lines 194-195: “Majority of taxa” is undefined. Please specify percent or number of taxa. “Several regions” is undefined. Specify how many regions, which ones they are, and discuss the importance of these regions with respect to crops, nematodes, etc. “Seems to vary” is undefined and subjective. Specify numerically or statistically the degree to which taxa vary, and what they vary with respect to (presence, abundance, etc).

Line 203: Please specify what economic issues are induced.

Line 204: Rather than “One of these two crops…”, crops should be specified.

Line 214: This entire sentence is vague. Specify soil conditions to which you refer, and how these collective soil conditions and/or abiotic factors influence PPN and free-living nematodes. Discuss how these collective soil and abiotic factors stimulate or repress nematode growth and development, and why this is important to your argument.

Line 216: “More or less susceptible” is vague. Explain how nematodes are susceptible to the presence of Zn and Cu in (presumably) soil.

Line 220: Do you mean “intoxicated” or nematodes that suffer from toxic effects?

Lines 222-223: Specify the “strong impact” due to heavy metals.

Line 228: “Major” is subjective; simply list the impact of soil texture on community structure quantitatively or statistically rather than rendering a judgment call as to perceived importance. Is the effect of statistical significance? Usage of “structuration” again.

Line 231: “Eased” is subjective. Please clarify.

Lines 235-237: Specify how rainfall affects nematode communities, how temperature affects PPNs, and why other soil variables are not considered important (and specify the soil variables). Why are all of these effects the case?

Line 239: Sentence appears to contradict itself, can this be made clearer?

Lines 242-243: “Soil work” and “structuration”—see notes above.

Lines 253-260: I’m having trouble following the flow of the argument; please clarify. Also, on line 256, the statement is made that the study has a large geographic scale, whereas in line 239 the opposite is stated. Please clarify.

Line 266: “certain groups of years”. Specify the dates and/or date ranges.

Line 271-272: “Their effects appeared to be positive or negative, probably depending on the PPN ecological niche.” This sentence is vague and requires considerable expansion; please discuss the specific positive and negative effects, and how these effects are or could be driven by a given ecological niche.

Line 273: “inputs”; please specify the inputs that you are referring to and how inputs are characterized to increase yield.

Line 274: “better quality” is subjective. What, specifically, constitutes quality, a change in quality, and what do these changes induce in nematode consortia?

CONCLUSIONS:

Lines 285, 287: In the discussion crop rotations are described as having been removed from analyses, therefore it should not be possible to draw conclusions from them. Please clarify.

Line 289: By “population” do you mean “abundance”?

MATERIALS AND METHODS:

Lines 317-318, 320: It is safe to say that many readers will not be familiar with the national monitoring program. Within the regions listed, please specify field site locations, including the geolocations of the sites. In line 320, specify which regions, sites, and fields were used and note where changes occurred between years.

Line 322: Specify the name of the agricultural data website, not just the reference. Were mean temperatures obtained for each field individually? If not, how were temperatures determined?

Line 333: What is an “elementary soil core”? This is not terminology that I am familiar with.

Line 344: Should this be “sieves”? If so, please specify all sieve sizes as well as any standard protocols used.

Line 345-346: “based on personal observations” is vague; specify the exact methodology or calculation required to determine the number of larvae per cyst.

Line 351: Please define “field surface”.

6. PLOS authors have the option to publish the peer review history of their article (what does this mean?). If published, this will include your full peer review and any attached files.

Reviewer #1: No

Reviewer #2: No

---

## [Author Response · Author response to Decision Letter 0]

3 Nov 2021

Response to the points raised by the academic editor and reviewers

Dear editor,

Please find here a response to reviewers’ comments and a substantially revised manuscript according to all the points raised. According to numerous comments from both reviewers, many sentences have been modified to remove subjective language, and clarifications have been made for table 1 and Methods section. As reviewers pointed out some misuse of different terms, supporting informations have also been modified to match the vocabulary in the manuscript.

Sincerely

Dr. Nathan Garcia

Editor comments:

_ Style and format have been checked and modified through the whole manuscript to meet PlosOne’s style requirements according to the templates from the links. Fig 1 has been uploaded to the PACE digital diagnostic tool as recommended.

_ This work was not peer-reviewed nor formally published. In fact, some of the results were obtained and presented in my PhD thesis, defended at the end of 2017, and the whole thesis manuscript is available on the HAL open archive (which is mandatory for French Ph.D. students), without any copyrights. Co-authors and I have continued working on this study since my PhD defense and thus the analyses and the results presented in this manuscript are not strictly the same as those from my Ph.D. thesis manuscript and most of the text has been rewritten. There is no reason to consider this submission as a dual publication of my PhD thesis manuscript. 

Reviewer #1 comments:

• Nematodes are not microorganisms, they belong to the mesofauna, please correct

_ Reviewer #2 also raised this point. The sentence has been removed from the abstract and corrected latter in the manuscript.

• To compete is not enough, they need to suppress the non-native species. Rephrase.

_ Ideally reviewer #1 is right, however we have few evidences of such suppression as in Garcia et al., 2018 (DOI: 10.1111/ppa.12914) we observed suppression of M. chitwoodi in only 5 pots out of 80 after a 4 months experiment. But abundance decrease has been observed much more frequently and we can expect that over a longer period this can lead to suppression or at least keep the non-native species at a non-damageable level. So to suppress is indeed the best solution but to compete is already a quite interesting observation as it can lead to suppression or some control over time. 

• “…using non-damaging indigenous species…” applies artificial application of those specimen, which is not realistic. I think what is meant is, that non-damaging species should be supported in order to suppress damaging species.

_ It is indeed what we meant. The sentence has been modified accordingly.

• Regarding the target organisms, I don’t think we can differ between non-native and native damaging species. Here we need a more general approach like “suppressing damaging species including non-native species.” Or similar.

_ We agree if the non-native species is already established in the soil. However, we placed this work in a scheme of an introduction of a species listed as quarantine organism that is not currently widely distributed in Europe and thus we consider that supporting indigenous non-damaging species may decrease the abundance of such introduced species.

• Looking of the spectrum of identified genera, they belong to more than one family. Rephrase accordingly.

_ Sentence has been rephrased. We identified 10 plant-parasitic nematode taxa, and for one of them, we were not able to identify the genus but only the family.

• Soils are part of fields. No need to mention separately.

_ Sentence has been modified. By “field” we meant “cultural practices” and by “soil” we meant “soil physico-chemical properties”.

• I don’t think data were not collected over six years, but instead field data representing the past six years were collected. Clarify.

_ Sentence has been modified. Indeed, we meant “representing the past six years”.

• … than what? Second partner of comparison needs to be mentioned.

_Sentence has been completed.

• Both aspects were not part of the study. Food quality refers to crop plant species grown and soil food web to the entire soil ecosystem. The presented data do not provide any information on the food quality of the crops grown nor on the characterization of the soil food web (suppressiveness etc). Change accordingly.

_ Sentence has been removed. We agree that the sentence was too general compared to the data analyzed in this work.

• Uncommen wording within this context. Do you mean establishment or spreading? Why managing and not controlling/eradicating/suppressing the introduced nematode? Rephrase accordingly.

_ “implantation” has been replaced by “establishment” all over the manuscript. This point was also raised by reviewer #2. To our opinion, “manage” is synonym to “controlling” suggested by reviewer #1 and as mentioned in a previous comment, we have only few evidence that “eradicating/suppressing” is possible, while abundance decrease is more obvious and already a useful lever. 

• Need to be mentioned, how this can be done. This is the most relevant aspect according to the authors, and thus the procedure should be described in detail. What are the options to promote diversity? How does the information gained from the collected data specifically help to promote diversity. Give at least suggestions.

_ Sentence has been removed since the reasoning was too indirect for the abstract for which we do not have space for more details. Here we meant that the variables highlighted in our work could positively influence PPN communities’ richness and abundance and thus promote biodiversity.

• See previous comment

_ “micro-organisms” replaced by “mesofauna”.

• No need to mention

_ Agree, text has been removed.

• What are natural population levels? Needs explanation.

_ Reviewer #2 also raised this point. “population level” has been modified by “abundance” all over the manuscript. By “natural abundance” we meant the abundance that is commonly observed in French fields, both according to our expertise but also to published data like Villenave et al. (2013) (DOI: 10.4236/ojss.2013.31005) in which they sampled and analyzed the nematode communities from 52 fields or Garcia et al. (2018) (DOI: 10.1163/15685411-00003136) where 72 fields were considered.

• establishment?

_ Yes, word has been replaced.

• See previous comment

_ Same.

• What do you mean by “competition between communities”? specify.

_ Sentence has been rephrased.

• I guess it is more the composition than the structure

_ As reviewer #2 suggested, we have replaced “structuration” by “community structure” for the whole manuscript.

• It is not clear to the reader how that information will improve national monitoring programs. A good understanding of the diversity and composition of nematode communities is highly complex and time consuming. I doubt that it will be realistic in the future to select fields in monitoring programs based on nematode diversity and composition. The main and most likely only factor is host plant status. Rephrase accordingly.

• As mentioned above, if you want to find M. chitwoodi, you sample good host plants at the end of the season (highest expected numbers) considering possible ways of entrance (e.g. fields were seed potatoes were planted). Composition of plant-parasitic nematodes is not an issue and I doubt it will be in the future.

_ Sentences pointed out by these two comments has been modified to make our reasoning clear. We totally agree that for national monitoring program for M. chitwoodi, it is essential to sample fields based on good host plants and we are not questioning this in the study. However, it is not humanly possible for the sanitary services to sample every fields with a good host plant in the country. Thus, we wanted with this work to promote additional elements that can be considered to choose in the most efficient way the fields to sample among those with a good host plant. As indigenous plant-parasitic nematode communities may decrease M. chitwoodi abundance, and as soil properties, cultural practices and climate may impact plant-parasitic nematode communities, we believe those should be relevant elements to consider in addition to the host plant. 

• Tillage per se can also enhance plant-parasitic nematode densities, such as minimum tillage. Thus it is more the intensive tillage including ploughing that has a negative impact on ppn. Please specify.

_ Sentence has been modified.

• “soil” indicates all nematodes, but non-plant parasitic nematodes are not covered in this study

_ Agree, the modification proposed by reviewer #1 has been accepted.

• Those numbers are quite high. Is this really juveniles or could it be cyst content (eggs + juveniles)?

_ It is an extrapolation of Heterodera cyst content.

More details have been added in Methods section according to a latter comment from reviewer #1 and a comment from reviewer #2

In a previous work published by Garcia et al. 2018 in Nematology (DOI 10.1163/15685411-00003136) we investigated the number of juveniles and eggs in Heterodera cysts sampled in 7 fields in Burgundy (France) during two consecutive years. A total of 33 cysts belonging to different species were individually crushed and the number of juveniles and eggs counted. The obtained mean juveniles + eggs per cyst was then used to convert cyst numbers of other French fields into juvenile and eggs numbers.

• See comment above

_ We removed “micro” as suggested

• composition?

_ “Structuration” has been replaced by “community structure” as suggested by reviewer #2

• Composition?

_ See previous comments.

• It is not obvious how this can be the case and thus needs explanation.

_ Sentence has been more detailed. We meant here that profound tillage modifies soil texture, destroy weeds and move the nematodes deeply in the soil. Since these tiny organisms have limited active migration abilities, profound tillage negatively affects food accessibility.

• Composition?

_ See previous comments.

• What kind of groups? Needs explanation

_ We detailed here the group of years mentioned in Table 2.

• Or alternatively, ….were lowest in soil with the lowest C/N ratio.

_ The suggested changes have been accepted.

• Pesticides in general? The survey only mentions herbicide. Stick to what has been studied and avoid generalizing things.

_ Sentence containing this term has been removed (see following comments from reviewer #2). However, to answer the question, here we meant pesticides in general (including herbicides). In the survey, farmers have indicated the number of herbicide application and other pesticides application (meaning fungicides, molluscicides and insecticides) that we called “non-herbicides” in the manuscript. As farmers are not always prone to give details about the other pesticides applications, we have had to stick to the sole total amount of non-herbicides application.

• are soil work, tillage and cultivation practices used synomously? If so, stick to one term (cultivation practices) and use this consistently. Change in entire text.

_ In the manuscript “cultivation practices” is more general and contain tillage practices, herbicide and non-herbicide applications and crop rotations. 

Furthermore, according to reviewer #2 comment, “soil work” have been replace by “tillage practices” in the entire manuscript and we have detailed “superficial” or “profound” each time.

• Mention the criteria for the selection of those fields

_ Details about the selection of the fields in the French monitoring program have been added.

• Too general as not all heavy metals were determined. Specifically mention those that have been considered in this study.

_ “Heavy metals” have been replaced by those considered in this work. 

• I don’t think there are techniques that distinguish between free-living and cyst nematodes. But there are techniques specifically for mobile stages and cyst stages. Is that what is meant? Needs clarification and precise wording.

_ Yes, we meant “mobile stages”, “free-living” has been replaced.

• Using the same technique, i.e. Oostenbrink elutriator? This does not make sense. Describe more preciously how extraction was done and mention a reference

_ The whole extraction paragraph has been substantially more detailed, according to the appropriate reference, the EPPO extraction protocols for nematodes. 

• Two times the same centrifugation? Needs explanation.

_ More details have been added. The first centrifugation is done with water and the second with MgSO4 with a density of 1.18.

• See previous comment

_ “free-living” has been removed from the modified sentence.

• I thought the nematodes are already extracted at this step. So, where does the soil extract comes from? Do you mean nematode suspension? Usually, nematodes are concentrated in a certain volume (e.g. 10 ml) and then an aliquot (e.g. 1 ml) is used to identify and count the specimen. Preciously describe the method used.

_ Yes we meant “nematode suspension”, the words have been replaced. More details about the dilution step have also been added. We counted the PPNs from 5mL of a diluted suspension (usually 100mL) depending of the abundance of nematode in the total suspension to be able to correctly identify and count them.

• …. or eggs plus juveniles per cyst? (= cyst content)

_ Yes we meant juveniles and eggs. Sentence has been modified accordingly.

• Difficult to understand how this was done. Generally, a defined number of cysts is crushed and cyst content in terms of number of eggs and juveniles determined and calculated back according to 100 g soil.

_ See previous comments for the detailed method. Information has been added.

• What is the message of Legendre and Gallagher? Explain

_ Details about the cited reference have been added at the end of the sentence.

• The number of references is far too long for a research article of this type. Keep in mind that this is not a review article. Condense number of references to 30-40. This can be easily done by focusing on the most relevant references and/or referring to review articles regarding general aspects.

_ We have reduced the number of cited reference by 13 but we preferred and strongly believe that it is best practice to refer to original publications than to reviews citing these original publications. Moreover, PlosOne guide for author does not limit the number of references.

Reviewer #2 comments:

• Line 44: Technically unicellular organisms are considered to be micro-organisms, whereas nematodes (multicellular) are considered microfauna or mesofauna. Please adjust this wording for clarity and consistency with the field.

_ See comment from reviewer #1.

• Line 47: Does “natural population levels” refer to crop population abundances or nematode abundances? Please specify.

_ “Population level” has been replaced by “abundance” in the whole manuscript according to reviewer comment.

• Lines 55-57: “…could be potential antagonists.” This sentence is unclear, and the implication is vague; please specify what the introduced species could be antagonists for or to, and how the antagonism might occur.

_ Sentence has been rephrased and an example of antagonism between indigenous PPN communities and introduced species is given in the following sentence.

• Line 70: The term “soil work” is a non-standard term that is used frequently throughout this manuscript in reference to cultivation practices. In the interests of clarity, please use the standard terms “tillage practices” or “cultivation practices”, and further detail as necessary.

_ “Soil work” has been modified in the manuscript and figures accordingly to this comment.

• Line 72: Please describe how cultivation and physico-chemical properties of the soil affects nematodes. When creating a line of reasoning it is not sufficient to simply state that there is an effect, the effect should be described.

_ Details have been added to the sentence.

• Line 73: Please list types of environmental conditions and crop practices that have been studied.

_ Sentence has been completed.

• Lines 73-74: Please describe the nature of restricted areas and limited experimental designs.

_ Sentence has been removed to avoid repetition with the previous and following ones.

• Line 78: Please remind us of the importance of a root crop vs. a cereal crop.

_ Details have been added and the beginning of the Methods section has also been more detailed according to a latter point raised by reviewer #2. 

• Lines 104, 106: Please specify whether per 100 g soil refers to 100 g wet (or field-moist) soil or dry soil?

_ We meant 100g of fresh soil. This information has been added.

• Lines 107-109: I did not see any information in the Methods about performing calculations for the Shannon Index. Please include methodology, reference(s), and data.

_ Those informations have been added to the Methods section. A line regarding Shannon index has been added to table 1.

• Lines 110-112. This sentence is unclear. What does this mean?: “…and Meloidogyne sp. mentioned in the article correspond to other species.”

_ In French legislation, it is mandatory to look for Meloidogyne chitwoodi and M. fallax if we found any Meloidogyne sp. individuals in a sample. As we found some Meloidogyne in several samples, we ensured that it was not one of the two quarantine species. Sentence has been modified to make it clear.

• Line 136: Misspelling of “this”. “…cumulative effect of few practices…”; please list the practices to which you refer. Also, it is mentioned that agricultural practices were not retained in the analysis, and the sentence immediately contradicts itself by stating that effects due to practices were observed. Could this entire sentence be clarified?

_ Sentence has been clarified. On Fig 1, only cultural practices of the sampling year are considered and none significantly affects the PPN communities (i.e. none appeared on the factorial map). However, when the cultural practices from the past years are also considered, effect of the successive non-herbicide applications and of the successive profound tillage appeared on the factorial maps presented in Fig S1.

• Line 138: Please specify what you mean by a field surface. Whether or not a result is difficult to explain it should be listed in the results; further explanation can be done in the discussion, and if indeed the explanation is due to analysis artifacts, further explanation of the nature of the artifact is needed.

_ Field surface is the field area, measured in hectare.

This result is still present in the Results section but we move the rest of the paragraph regarding a possible artefact in the Discussion section with clarifications of our reasoning (line 271-277 in the track changes version).

• Lines 139-140: The methodology as described in this sentence appears to match sampling protocols outlined in the Methods section, thus contradicting the opening of the sentence. Please describe in detail what was not adaptable for this particular sampling.

_ Here we meant that we did not sample more soil in largest fields. Sentence has been modified to clarify.

• Line 143: Please reference the appropriate figure or table after “…factorial map.”

_ Reference to the Fig 1 has been added.

• Line 144: “seem” is subjective; simply list the impact of the variables quantitatively or statistically rather than rendering a judgment call as to perceived importance. Commentary of that nature should be moved to the discussion.

_ Sentence has been modified to remove “seem to”.

• Line 144: By “structuration” do you mean “community structure”? Structuration is not the standardly accepted term for community structure, nor is it commonly used to refer to the formation or change of community structure.

_ Yes, we meant “community structure”. The word has been modified for the whole manuscript.

• Lines 167-182: The definitive statements “None of these variables impacted…”, “…had a positive impact…”, “…had a negative impact…” cannot be made. Remember that this study is largely statistical, and that that metadata originates from soil data that has not been measured in the laboratory for this study. At most it is possible to state that “…have appeared to positively/negatively impact…”. This paragraph requires significant clarification.

_ The whole paragraph has been modified to remove definitive statements about soil variables (lines 183-200 of the track changes version). We agree that the soil data were not measured during this work but we have minimized this bias as much as possible by using the database developed by the French National Research Institute for Agriculture, Food, and Environment. Data are available at the canton scale (the smallest French administrative scale with a mean surface of 192Km²). For each canton, more than 10 soil samples were analyzed over the canton surface between 2010 and 2014. As the samples for this work took place in 2015 and 2016, we considered that those data were close enough to our conditions. 

• Lines 187-188: “Various” is vague usage. Please specify the geographical areas and crop management approaches to which you refer.

_ Sentence has been clarified. This first sentence was general to highlight that each field presented different environmental conditions (climatic and soil) and different cultural practices (for example from 0 to 6 profound tillage for the past 6 years).

• Line 189: What do you mean by “close to that of other similar agrosystems”? “Close” is subjective; what of your data supports this assertion, and please be very specific in your response. What are “similar” agrosystems?

_ Sentence has been clarified. We meant agrosystems that also include cereals and vegetables in the crop rotation similarly to the fields sampled in our study.

• Lines 194-195: “Majority of taxa” is undefined. Please specify percent or number of taxa. “Several regions” is undefined. Specify how many regions, which ones they are, and discuss the importance of these regions with respect to crops, nematodes, etc. “Seems to vary” is undefined and subjective. Specify numerically or statistically the degree to which taxa vary, and what they vary with respect to (presence, abundance, etc).

_ Sentence has been modified to give numbers of taxa and regions. We remove subjective end of the sentence, which is discussed in the following sentence.

• Line 203: Please specify what economic issues are induced.

_ We specify the economic issues from the cited references. 

• Line 204: Rather than “One of these two crops…”, crops should be specified.

_ Crops (beet or carrot) mentioned in the previous sentence have been specified here.

• Line 214: This entire sentence is vague. Specify soil conditions to which you refer, and how these collective soil conditions and/or abiotic factors influence PPN and free-living nematodes. Discuss how these collective soil and abiotic factors stimulate or repress nematode growth and development, and why this is important to your argument.

_ Sentence has been modified to be more specific about the abiotic factors and the discussion about their impact is detailed during the rest of the paragraph.

• Line 216: “More or less susceptible” is vague. Explain how nematodes are susceptible to the presence of Zn and Cu in (presumably) soil.

_ Sentence have been detailed with information from the cited reference.

• Line 220: Do you mean “intoxicated” or nematodes that suffer from toxic effects?

_ Yes we meant “intoxicated”. Sentence has been slightly modified to make it more understandable.

• Lines 222-223: Specify the “strong impact” due to heavy metals.

_ Sentence has been clarified in order to develop the reasoning.

• Line 228: “Major” is subjective; simply list the impact of soil texture on community structure quantitatively or statistically rather than rendering a judgment call as to perceived importance. Is the effect of statistical significance? Usage of “structuration” again.

_ We removed the word “major” and added information about the kind of impact of soil texture. Furthermore, more details are given in the following sentences.

• Line 231: “Eased” is subjective. Please clarify.

_ Sentence has been clarified to explain that root exudate are used by nematodes to locate their host plants.

• Lines 235-237: Specify how rainfall affects nematode communities, how temperature affects PPNs, and why other soil variables are not considered important (and specify the soil variables). Why are all of these effects the case?

_ These sentences have been reformulated and more details given about the effects of rainfalls and temperature observed in the cited reference.

• Line 239: Sentence appears to contradict itself, can this be made clearer?

_ Sentence has been clarified. Here we meant that the French climate is similar from north to south regarding the Köppen-Geiger climate classification map.

• Lines 242-243: “Soil work” and “structuration”—see notes above.

_ See previous comments.

• Lines 253-260: I’m having trouble following the flow of the argument; please clarify. Also, on line 256, the statement is made that the study has a large geographic scale, whereas in line 239 the opposite is stated. Please clarify.

_ Paragraph has been detailed to clarify our reasoning. Concerning the geographic scale, we meant here that at the country scale of this work, environmental variables have more impact on communities’ structure than the crop rotations that are very similar across the sampled fields (majority of cereals). Line 239 on the contrary, we stated that France is not a large country compare to the worldwide scale studied in the cited reference (Nielsen et al. (2014)) to highlight that climatic variations in our work are less diverse than climatic variations at a worldwide scale.

• Line 266: “certain groups of years”. Specify the dates and/or date ranges.

_ See reviewer #1 comment.

• Line 271-272: “Their effects appeared to be positive or negative, probably depending on the PPN ecological niche.” This sentence is vague and requires considerable expansion; please discuss the specific positive and negative effects, and how these effects are or could be driven by a given ecological niche.

_ Sentence has been removed as discussion about soil physico-chemical variables impacts on PPN taxa occur in the following sentences.

• Line 273: “inputs”; please specify the inputs that you are referring to and how inputs are characterized to increase yield.

_ Sentence has been slightly modified. We meant here “fertilizers” that are used to enrich the soils for better plant growth.

• Line 274: “better quality” is subjective. What, specifically, constitutes quality, a change in quality, and what do these changes induce in nematode consortia?

_ “better quality” has been removed and we have detailed what we meant here. Positive effect of fertilization on PPN abundance may be due to more dense root systems that will allow a higher PPN development as the resource is abundant.

• Lines 285, 287: In the discussion crop rotations are described as having been removed from analyses, therefore it should not be possible to draw conclusions from them. Please clarify.

_ First sentence of the Conclusions section has been removed as it is now more detailed in the discussion about the crop rotations (see previous comments).

• Line 289: By “population” do you mean “abundance”?

_ Yes. The word “population” has been consequently modified by “abundance”.

• Lines 317-318, 320: It is safe to say that many readers will not be familiar with the national monitoring program. Within the regions listed, please specify field site locations, including the geolocations of the sites. In line 320, specify which regions, sites, and fields were used and note where changes occurred between years.

_ According to reviewer #1 comment, details about the French monitoring program for M. chitwoodi and M. fallax have been added.

It is not possible to give more details about the geolocation of the fields as this information is confidential for the French phytosanitary services and farmers agreed to participate to this work only on an anonymously basis.

As requested, informations were added to highlight where changes occurred between years (lines 366-367 of the track change version).

• Line 322: Specify the name of the agricultural data website, not just the reference. Were mean temperatures obtained for each field individually? If not, how were temperatures determined?

_ We specified the name of the website and gave more details about the data collections. Data were collected for each field individually from the closest meteorological station.

• Line 333: What is an “elementary soil core”? This is not terminology that I am familiar with.

_ In this work, it is a soil carrot over the first 30 centimeters of soil.

• Line 344: Should this be “sieves”? If so, please specify all sieve sizes as well as any standard protocols used.

_ More information has been added about the sieves and the whole extraction processes, following other comments from reviewer #1

• Line 345-346: “based on personal observations” is vague; specify the exact methodology or calculation required to determine the number of larvae per cyst.

_ In a previous work published by Garcia et al. 2018 in Nematology (DOI 10.1163/15685411-00003136) we investigated the number of juveniles and eggs in Heterodera cysts sampled in 7 fields in Burgundy during two consecutive years. A total of 33 cysts belonging to different species were individually crushed and the number of juveniles and eggs counted. The obtained mean juveniles + eggs per cyst was then used to convert cyst numbers of other French fields into juvenile and eggs numbers.

• Line 351: Please define “field surface”.

_ See previous comment.

---

## [Decision Letter · Decision Letter 1]

6 Jan 2022

PONE-D-21-19128R1Diversity of plant parasitic nematodes characterized from fields of the French national monitoring programme for the Columbia root-knot nematodePLOS ONE

Dear Dr. Garcia,

Thank you for submitting your manuscript to PLOS ONE. After careful consideration, we feel that it has merit but does not fully meet PLOS ONE’s publication criteria as it currently stands. Therefore, we invite you to submit a revised version of the manuscript that addresses the points raised during the review process.

The manuscript has been reviewed by two experts, both of which agree that you have adequately addressed the points raised during the first review round.

Please consider the comments of the reviewers carefully; I agree with Reviewer 3 that the inclusion of supporting metadata from fields that are indeed infested with M. chitwoodi could potentially strengthen the discussion and value of your results. I acknowledge that your study was initially designed to identify parameters that could aid the development of a focused M. chitwoodi monitoring scheme, but as you (fortunately for the owners of the sampled fields!) did not detect the species, you could consider putting less emphasis on this objective. As we are far from understanding which parameters/variables are key to the distribution of nematode taxa in general, the results on the plant parasitic nematode community is valuable on its own, even though you did not find M. chitwoodi.

Below, you will find my specific comments; please revise your manuscript accordingly:

L. 29: ”anthropic … variables” sounds a bit odd; please rephrase to “cultivation practices and environmental variables”.

Throughout manuscript: Do not use “anthropic”, please change to “anthropogenic” or, if appropriate, use “cultivation practice”, which more accurately describes the nature of anthropogenic activities.

L. 35: Again, there is no such thing as “anthropic soil variables”.

L. 44: In soil, I would claim that nematodes belong to the microfauna (as defined as metazoan with a body width ≤100 µm). If you classify according to body length, nematodes belong to the mesofauna. However, the body width is much more relevant, as it determines which soil pore opening are accessible to the nematodes.

L. 47: Like reviewer 1, I am also puzzled by the “natural abundances”; what would define “unnatural abundances” then? Better to state that most PPNs are not damaging to crops at the abundances commonly found in French agricultural soils (or something equivalent).

Table 1: Provide more information in the table legend; e.g. that data depict the mean of 35 and 31 fields surveyed in 2015 and 2016, respectively, and clarify what “Prevalence” and “W” refer to.

Fig. 1: In l. 123-125, you mention that Fig. 1 includes the cultural practices, but I only see environmental variables, no reference to cultivation practices. Please clarify.

Table 2: Please explain the variables (C/N, Herbi, Zn) in the legend. The meaning of “Year period” should also be explained. Also describe which type of model the table depict; in essence, the table and figures should be self-explanatory.

L. 210-211: Which better host plant?

L. 215-236: You do not discuss your own results here; please do so.

L. 216-226: Are the referenced studies on heavy metal effects conducted in soil or in other matrices/media? Generally, the bioavailability and negative effects of heavy metals on soil organisms are dramatically reduced in soil compared to e.g. liquid media, so this information is quite important to evaluate the comparability to your study.

L. 218: Please add a reference that shows negative correlation between Zn conc. and PPN abundance.

L. 245: Coming from a small country, I find that France is a rather large country. In any case, the important issue here is not the size of the country, but climatic variation, so I suggest you omit reference to the size of the country and focus on climatic variation across the sampled regions.

L. 249: “field surface area”.

L.250-51: “… we did not sample more samples in the larger fields), but sampled seven soil core over the longest diagonal in all fields”

L. 252: “surface area affected PPN communities”

L. 269: “contrasts”

L. 368: “surface area”

L. 388: Please define “MCA”

L. 402 and 403:”surface area”

L. 405: Define “SW” and “CI”

We look forward to receiving your revised manuscript.

Kind regards,

Mette Vestergård, Ph.D.

Academic Editor

PLOS ONE

Additional Editor Comments:

The manuscript has been reviewed by two experts, both of which agree that you have adequately addressed the points raised during the first review round.

Please consider the comments of the reviewers carefully; I agree with Reviewer 3 that the inclusion of supporting metadata from fields that are indeed infested with M. chitwoodi could potentially strengthen the discussion and value of your results. I acknowledge that your study was initially designed to identify parameters that could aid the development of a focused M. chitwoodi monitoring scheme, but as you (fortunately for the owners of the sampled fields!) did not detect the species, you could consider putting less emphasis on this objective. As we are far from understanding which parameters/variables are key to the distribution of nematode taxa in general, the results on the plant parasitic nematode community is valuable on its own, even though you did not find M. chitwoodi.

Below, you will find my specific comments; please revise your manuscript accordingly:

L. 29: ”anthropic … variables” sounds a bit odd; please rephrase to “cultivation practices and environmental variables”.

Throughout manuscript: Do not use “anthropic”, please change to “anthropogenic” or, if appropriate, use “cultivation practice”, which more accurately describes the nature of anthropogenic activities.

L. 35: Again, there is no such thing as “anthropic soil variables”.

L. 44: In soil, I would claim that nematodes belong to the microfauna (as defined as metazoan with a body width ≤100 µm). If you classify according to body length, nematodes belong to the mesofauna. However, the body width is much more relevant, as it determines which soil pore opening are accessible to the nematodes.

L. 47: Like reviewer 1, I am also puzzled by the “natural abundances”; what would define “unnatural abundances” then? Better to state that most PPNs are not damaging to crops at the abundances commonly found in French agricultural soils (or something equivalent).

Table 1: Provide more information in the table legend; e.g. that data depict the mean of 35 and 31 fields surveyed in 2015 and 2016, respectively, and clarify what “Prevalence” and “W” refer to.

Fig. 1: In l. 123-125, you mention that Fig. 1 includes the cultural practices, but I only see environmental variables, no reference to cultivation practices. Please clarify.

Table 2: Please explain the variables (C/N, Herbi, Zn) in the legend. The meaning of “Year period” should also be explained. Also describe which type of model the table depict; in essence, the table and figures should be self-explanatory.

L. 210-211: Which better host plant?

L. 215-236: You do not discuss your own results here; please do so.

L. 216-226: Are the referenced studies on heavy metal effects conducted in soil or in other matrices/media? Generally, the bioavailability and negative effects of heavy metals on soil organisms are dramatically reduced in soil compared to e.g. liquid media, so this information is quite important to evaluate the comparability to your study.

L. 218: Please add a reference that shows negative correlation between Zn conc. and PPN abundance.

L. 245: Coming from a small country, I find that France is a rather large country. In any case, the important issue here is not the size of the country, but climatic variation, so I suggest you omit reference to the size of the country and focus on climatic variation across the sampled regions.

L. 249: “field surface area”.

L.250-51: “… we did not sample more samples in the larger fields), but sampled seven soil core over the longest diagonal in all fields”

L. 252: “surface area affected PPN communities”

L. 269: “contrasts”

L. 368: “surface area”

L. 388: Please define “MCA”

L. 402 and 403:”surface area”

L. 405: Define “SW” and “CI”

Reviewers' comments:

Reviewer's Responses to Questions

**Comments to the Author**

1. If the authors have adequately addressed your comments raised in a previous round of review and you feel that this manuscript is now acceptable for publication, you may indicate that here to bypass the “Comments to the Author” section, enter your conflict of interest statement in the “Confidential to Editor” section, and submit your "Accept" recommendation.

Reviewer #2: All comments have been addressed

Reviewer #3: (No Response)

2. Is the manuscript technically sound, and do the data support the conclusions?

Reviewer #2: Yes

Reviewer #3: Yes

3. Has the statistical analysis been performed appropriately and rigorously? 

Reviewer #2: Yes

Reviewer #3: Yes

4. Have the authors made all data underlying the findings in their manuscript fully available?

Reviewer #2: Yes

Reviewer #3: Yes

5. Is the manuscript presented in an intelligible fashion and written in standard English?

Reviewer #2: Yes

Reviewer #3: Yes

6. Review Comments to the Author

Reviewer #2: All comments have been addressed, and I appreciate the authors' thoroughness in doing so; considerable clarity has been added. I would suggest a final check of the manuscript to clean up any minor grammar issues and/or misused plurals, but aside from this I recommend publication of this manuscript.

Reviewer #3: Overall the authors took into consideration the major points raised by the reviewers and improved the manuscript substantially.

In the abstract, introduction and discussion the authors emphasize that the obtained knowledge described in this work can facilitate management of the establishment of an introduced species (M. chitwoodi). Basically the aim is to look for parameters to develop an more focussed sampling scheme to monitor M. chitwoodi/M. fallax infestations. Unfortunately in none of the sampled fields M. chitwoodi was detected. It would have been an added value if M. chitwoodi-infested fields were included in this study. If metadata are available for known infested fields it would be useful to compare these with the results from this study and add this comparison to the discussion. With the provided data no conclusion can be made regarding higher possibilities for establishment of M. chitwoodi.

Statement in line 49-51 is not backed by references given, other refs should be used.

7. PLOS authors have the option to publish the peer review history of their article (what does this mean?). If published, this will include your full peer review and any attached files.

Reviewer #2: No

Reviewer #3: No

---

## [Author Response · Author response to Decision Letter 1]

19 Feb 2022

Response to the points raised by the academic editor and reviewers

Dear editor,

Please find here a response to reviewer 3 and your comments and a revised manuscript according to all the points raised. All the comments have been addressed except the inclusion of metadata regarding PPN communities in fields where M. chitwoodi is present (see following comment) as we did not find such relevant information in the scientific literature. However, we have slightly modified the text to consider those comments.

Sincerely

Dr. Nathan Garcia

Editor comments:

• I agree with Reviewer 3 that the inclusion of supporting metadata from fields that are indeed infested with M. chitwoodi could potentially strengthen the discussion and value of your results. I acknowledge that your study was initially designed to identify parameters that could aid the development of a focused M. chitwoodi monitoring scheme, but as you (fortunately for the owners of the sampled fields!) did not detect the species, you could consider putting less emphasis on this objective. As we are far from understanding which parameters/variables are key to the distribution of nematode taxa in general, the results on the plant parasitic nematode community is valuable on its own, even though you did not find M. chitwoodi.

_ We agree that metadata about nematode communities in fields infested with M. chitwoodi would be an interesting implementation in this study. However, we did not find any relevant publication in the literature to do so since we only found publications focusing on experimental design instead of agricultural fields. That appear not suitable for comparison with our study since the experiments in those studies had a major impact on the nematode communities and M. chitwoodi abundance prior to any other parameter. Furthermore as M. chitwoodi is regulated in many countries, fields that are known to be infested are usually under specific management that depend on the countries legislation. These managements implies that the whole nematode communities are drastically impacted as there is no specific management targeting M. chitwoodi alone. Thus, to our opinion, even those data would not be very relevant to compare with our work because they would not displayed the nematode communities commonly found in agricultural soil as we are doing in this study. 

However, we have followed your suggestion to put less emphasis on M. chitwoodi in the revised version of the manuscript as we totally agree that impact of the studied variables on PPN communities is valuable on its own. We still believe that this kind of study can help the phytosanitary services to target the fields in their monitoring scheme as we know that PPN communities have an impact on M. chitwoodi abundance (since we observed it in Garcia et al. 2018 (DOI: 10.1111/ppa.12914)). But as you mentioned, we are far from understanding the whole mechanisms of soil nematode communities distribution. Thus several sentences of the abstract and introduction have been removed or modified.

• L. 29: ”anthropic … variables” sounds a bit odd; please rephrase to “cultivation practices and environmental variables”.

_ Agree, the sentence has been modified and according to the following comments, “anthropic” has been replace by the appropriate term for the whole manuscript. 

• Throughout manuscript: Do not use “anthropic”, please change to “anthropogenic” or, if appropriate, use “cultivation practice”, which more accurately describes the nature of anthropogenic activities.

_ See previous comment

• L. 35: Again, there is no such thing as “anthropic soil variables”.

_ See previous comment. Here we meant “the cultivation practices that have an impact on soil” (such as tillage).

• L. 44: In soil, I would claim that nematodes belong to the microfauna (as defined as metazoan with a body width ≤100 µm). If you classify according to body length, nematodes belong to the mesofauna. However, the body width is much more relevant, as it determines which soil pore opening are accessible to the nematodes.

_ We agree but both reviewers 1 and 2 considered that nematodes belong to mesofauna. We have remove this terms from the sentence.

• L. 47: Like reviewer 1, I am also puzzled by the “natural abundances”; what would define “unnatural abundances” then? Better to state that most PPNs are not damaging to crops at the abundances commonly found in French agricultural soils (or something equivalent).

_ Agree, this is exactly what we meant here and the sentence has been modified accordingly.

• Table 1: Provide more information in the table legend; e.g. that data depict the mean of 35 and 31 fields surveyed in 2015 and 2016, respectively, and clarify what “Prevalence” and “W” refer to.

_ Table legend has been detailed accordingly.

• Fig. 1: In l. 123-125, you mention that Fig. 1 includes the cultural practices, but I only see environmental variables, no reference to cultivation practices. Please clarify.

_ Here we meant that, regardless the outcome of the factorial analysis (i.e. the factorial map presented in Fig.1), cultural practices were implemented to run the analysis and that for fig. 1, only the cultural practices of the sampling year were implemented (not those of the past years that are implemented to run the analyses shown in S1Fig.). As tb-RDA allow a selection step thanks to a permutation test, Fig.1 display only the significant variables which are, as you mention, only environmental variables. 

The text has been modified in order to clarify.

• Table 2: Please explain the variables (C/N, Herbi, Zn) in the legend. The meaning of “Year period” should also be explained. Also describe which type of model the table depict; in essence, the table and figures should be self-explanatory.

_ Legend has been substantially detailed and the model type (i.e. Poisson GLMM) has been added in title.

• L. 210-211: Which better host plant?

_ It is impossible to say as species of this family feeds on a very large host range, including cereals that have mainly been grown during 2016 for the sampled fields, but also many weeds that have not been considered in this study. However, we have slightly detailed what we meant in the sentence.

• L. 215-236: You do not discuss your own results here; please do so.

_ Agree. The whole paragraph has been detailed to link the literature knowledge to our own results.

• L. 216-226: Are the referenced studies on heavy metal effects conducted in soil or in other matrices/media? Generally, the bioavailability and negative effects of heavy metals on soil organisms are dramatically reduced in soil compared to e.g. liquid media, so this information is quite important to evaluate the comparability to your study.

_ References 16, 33, 34 studied heavy metal effect on nematode communities in soil and references 31 and 32 studied their effect on C. elegans on a growing media but observed detrimental effect even at low concentration. In our discussion, references 16, 33 and 34 are cited to be compared to our results and while reference 31 and 32 are cited to explain the physiological mechanisms involved. As the paragraph has been modified accordingly to the previous comment, we also included these elements.

• L. 218: Please add a reference that shows negative correlation between Zn conc. and PPN abundance.

_ L.218, we were discussing about our own results. We agree that it was not clear and thus the sentence has been modified.

• L. 245: Coming from a small country, I find that France is a rather large country. In any case, the important issue here is not the size of the country, but climatic variation, so I suggest you omit reference to the size of the country and focus on climatic variation across the sampled regions.

_ Agree, and this is what we meant. Reference to the country size has been removed in the text.

• L. 249: “field surface area”.

_ Sentence has been replaced

• L.250-51: “… we did not sample more samples in the larger fields), but sampled seven soil core over the longest diagonal in all fields”

_ Sentence has been replaced

• L. 252: “surface area affected PPN communities”

_ Sentence has been replaced

• L. 269: “contrasts”

_ Corrected

• L. 368: “surface area”

_ Sentence has been modified

• L. 388: Please define “MCA”

_ MCA (for multiple correspondence analysis) has been defined

• L. 402 and 403:”surface area”

_ Sentences has been modified 

• L. 405: Define “SW” and “CI”

_ SW (for sum of weight) and CI (for confident interval) have been defined.

Reviewer #3 comments:

• In the abstract, introduction and discussion the authors emphasize that the obtained knowledge described in this work can facilitate management of the establishment of an introduced species (M. chitwoodi). Basically the aim is to look for parameters to develop an more focussed sampling scheme to monitor M. chitwoodi/M. fallax infestations. Unfortunately in none of the sampled fields M. chitwoodi was detected. It would have been an added value if M. chitwoodi-infested fields were included in this study. If metadata are available for known infested fields it would be useful to compare these with the results from this study and add this comparison to the discussion. With the provided data no conclusion can be made regarding higher possibilities for establishment of M. chitwoodi.

_ See first comment from editor.

• Statement in line 49-51 is not backed by references given, other refs should be used 

_ References used here has been modified.

---

## [Editor Report · Decision Letter 2]

23 Feb 2022

Diversity of plant parasitic nematodes characterized from fields of the French national monitoring programme for the Columbia root-knot nematode

PONE-D-21-19128R2

Dear Dr. Garcia,

We’re pleased to inform you that your manuscript has been judged scientifically suitable for publication and will be formally accepted for publication once it meets all outstanding technical requirements.

Kind regards,

Mette Vestergård, Ph.D.

Academic Editor

PLOS ONE

Additional Editor Comments (optional):

Reading through the manuscript I noticed some errors listed below that you should amend when you do the final review of the manuscript:

L. 111: ”These”

L. 191: ”the presence of the same PPN…”

L. 251: “cores”

L. 301: There is no such thing as a “”PPN ecosystem”; please rephrase – I presume you mean “PPN communities”
---

## [Editor Report · Acceptance letter]

28 Feb 2022

PONE-D-21-19128R2 

Diversity of plant parasitic nematodes characterized from fields of the French national monitoring programme for the Columbia root-knot nematode 

Dear Dr. Garcia:

I'm pleased to inform you that your manuscript has been deemed suitable for publication in PLOS ONE. Congratulations! Your manuscript is now with our production department. 

Kind regards, 

on behalf of

Dr. Mette Vestergård 

Academic Editor

PLOS ONE